# A Comprehensive Methodology for Evaluating the Economic Impacts of Floods: An Application to Canada, Mexico, and the United States

Xin Wen [1,2,*], Ana María Alarcón Ferreira [3,4], Lynn M. Rae [5], Hirmand Saffari [1], Zafar Adeel [1], Laura A. Bakkensen [6], Karla M. Méndez Estrada [7], Gregg M. Garfin [5], Renee A. McPherson [8] and Ernesto Franco Vargas [7]

1   Pacific Water Research Centre, Simon Fraser University, Surrey, BC V3T 0A3, Canada
2   Department of Biology and Institute of Environmental and Interdisciplinary Science, Carleton University, Ottawa, ON K1S 5B6, Canada
3   Posgrado en Ciencias de la Tierra, Universidad Nacional Autónoma de México (UNAM), Mexico City 04510, Mexico
4   Licenciatura de Protección Civil y Gestión de Riesgos, Universidad Autónoma de la Ciudad de México (UACM/SLT), Mexico City 06720, Mexico
5   School of Natural Resources and the Environment, University of Arizona, Tucson, AZ 85721, USA
6   School of Government & Public Policy, University of Arizona, Tucson, AZ 85721, USA
7   Centro Nacional de Prevención de Desastres, Mexico City 04360, Mexico
8   South Central Climate Adaptation Science Center, Department of Geography and Environmental Sustainability, University of Oklahoma, Norman, OK 73019, USA
*   Correspondence: xinwen4@cunet.carleton.ca

**Abstract:** In 2020, we developed a comprehensive methodology (henceforth, the methodology) to assess flood-related economic costs. The methodology covers direct damages, indirect effects, and *losses and additional costs* across 105 social, infrastructure, economic, and emergency response indicators. As a companion paper, this study presents findings from analysis of applying the methodology to investigate economic costs for major flood events between 2013 and 2017 and to assess gaps in the existing datasets across Canada, Mexico, and the United States. In addition, we conducted one case study from each country for an in-depth examination of the applicability of the methodology. Applying the methodology, Mexico showed the most complete flood indicator data availability and accessibility among the three countries. We found that most flood-related economic cost assessments evaluated only direct damages, and indirect effect data were rarely included in datasets in the three countries. Moreover, few of the records from Canada and the United States captured the *losses* and *additional costs*. Flood-related economic cost data at the municipal or county level were easily accessible in Mexico and the United States. Mexico's National Center for Prevention of Disasters (Centro Nacional de Prevención de Desastres), unique among the three nations, provided access to centralized and comprehensive flood cost data. In the United States and Canada, data collection by multiple agencies that focus on different jurisdictions and scales of flood damage complicated comprehensive data collection and led to incomplete economic cost assessments. Our analysis strongly suggests that countries should aim to expand the set of data indicators available and become more granular across space and time while maintaining data quality. This study provides significant insights about approaches for collating spatial, temporal, and outcome-specific localized flood economic costs and the major data gaps across the three countries.

**Keywords:** flood economic impact; tri-national assessment; data accessibility and availability

## 1. Introduction

Floods are among the most common and destructive disasters in the world [1–4]. Moreover, many recent studies highlight that climate change has influenced water-related variables (e.g., water vapor content, rainfall, and snowmelt) that contribute to floods, likely

leading to increased frequency and intensity of floods [5–8]. Such losses and damages have prompted a global disaster risk reduction agenda and international cooperation. For example, the United Nations Office for Disaster Risk Reduction (UNDRR) released the Sendai Framework for disaster risk reduction 2015–2030 in 2015 to guide nations in reducing the risk of negative consequences of disasters [9]. The 2022 Global Platform took stock of the implementation of this framework and highlighted that a comprehensive disaster and climate risk management method plays a key role in disaster risk reduction [10]. In particular, the platform highlighted the need to strengthen data ecosystems through cross-national cooperation by increasing the access, ease of use, and synthesis of many types of data generated throughout the disaster process to better inform and achieve risk-reduction goals.

In North America, flood damages have recently become the principal source of property insurance claims in Canada [11]. Floods caused by thunderstorms and hurricanes generate substantial economic losses and destruction of infrastructure and property each year in Mexico, especially areas along the coasts of the Pacific Ocean and Gulf of Mexico [12]. Similarly, floods in the United States have caused billions of dollars of damages over the past three decades [13]. Consequently, significant policy attention has been focused on developing effective flood risk governance across these countries [12,14–17]. Despite the persistent and sizable adverse impacts, these countries do not gather and record flood-impact-related economic data consistently and thoroughly, making integrated and coordinated efforts to address flood risk across North America difficult [1].

To fill this important gap, Adeel et al. [1] reviewed methods for estimating the economic costs of flood damages across Canada, Mexico, and the United States. The paper documented inconsistency in the approaches across the three countries for measuring the costs of flood damages, and approaches were designed to achieve different project objectives. Likewise, assessing the economic impacts of floods was often incomplete because most loss assessments were performed separately per sector [1]. As a result, Adeel et al. [1] offered a comprehensive flood cost assessment methodology (henceforth, the methodology) to support disaster response and flood risk management policies in the three countries. The methodology covers direct damages (direct damage occurs immediately or within a few hours of its occurrence, e.g., dwelling damage [1]), indirect effects (indirect effect is related to second-order effects due to flooding on products and housing markets, etc. [1]), and *losses and additional costs* (*loss and additional costs* are disruptions to flows resulting from a flood, e.g., temporary accommodation [1]) across 105 social, infrastructure, economic, and emergency response indicators. It embraced guiding principles of the Sendai Framework, including coherence of risk reduction practices across different sectors, meaningful and strong international cooperation, and accounting for local characteristics of disaster risk.

This study is a companion paper to Adeel et al. [1], which solely focused on the development of the methodology. Here, we test the methodology by analyzing existing flood economic cost data in the three countries and offer insights into any major data gaps we found. The objectives of this study are twofold: First, to apply and summarize the findings of the methodology in the three nations from a five-year test window (2013–2017), including three in-depth case studies. Second, to discuss successes, challenges, and actions needed to improve the current ways the nations collect and manage data on flood damages and losses. Our major flood-event analysis builds on existing data to examine which economic sectors have the most complete data availability and accessibility. Our case studies offer insights into data inconsistencies among data sources. In addition, our detailed flood cost analysis is crucial for planning strategic investments in communities and infrastructure that will build resilience against future events.

## 2. Methodology

### 2.1. A Brief Recap of the Methodology Development

We begin with a brief recap of the flood costing methodology presented in Adeel et al. [1]. The Commission for Environmental Cooperation (The Commission for Envi-

ronmental Cooperation—established in 1994 through the North American Agreement on Environmental Cooperation—facilitates collaboration and public participation to foster conservation, protection, and enhancement of the North American environment for the benefit of present and future generations, in the context of increasing economic, trade, and social links among Canada, Mexico, and the United States.) (CEC) project, entitled "Costing Floods and Other Extreme Events," commenced in May 2019. Our tri-national team first conducted a systematic review of approaches used to assess the flood economic costs in Canada, Mexico, and the United States. In September 2019, the first CEC project workshop was held in Vancouver, Canada, and included participants from government, academia, and the insurance industry. Based on this review, the team and workshop participants agreed that the United Nations Economic Commission for Latin America and the Caribbean (UN-ECLAC) methodology [18] could be further enhanced to cover more sectors and aspects. UN-ECLAC's methodology is a macro-economic assessment of damage, losses, and additional costs that result from a disaster and primarily has focused on major economic sectors in Latin America (e.g., agriculture, manufacturing, tourism). A version adapted by Bitrán [19] has been used in Mexico for economic assessment impact for earthquakes, volcanoes, floods, and other disasters since 2005. Adeel et al. [1] enhanced the UN-ECLAC methodology in several sectoral categories, particularly the social, infrastructure, and transportation sectors.

Figure 1 shows the key elements of the proposed methodology. Please refer to Adeel et al. [1] for details of the first CEC project workshop and flood cost indicators. The authors discussed and amended the methodology further after an Indigenous perspective workshop (July 2020), and the academic peer-review process. Indigenous knowledge offered a broader perspective on how cultural resources could be included in the methodology (e.g., erosion and sedimentation and wildlife and aquatic species health). The first CEC project workshop also recommended that the methodology be tested using real-world data to examine its applicability and performance and suggested the five-year time period (2013–2017) for testing the proposed methodology. This five-year window ensured that the nations had sufficient time for data collection and input, such as identifying where, when, and how the flooding impacts had occurred [1]. The project budget limited the period to five years, as collection of these secondary data was time consuming.

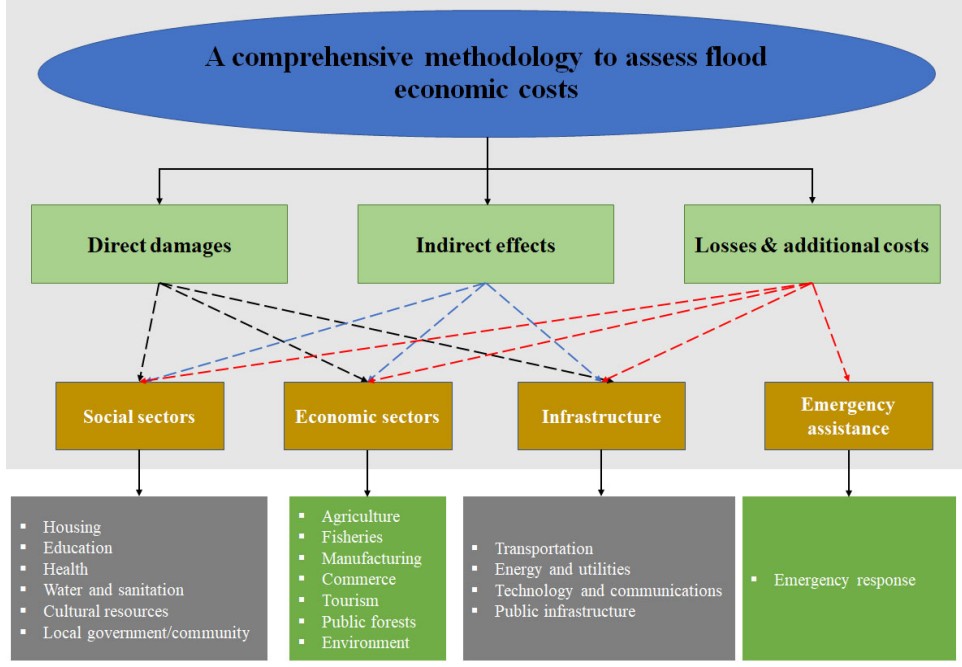

**Figure 1.** Key elements of the methodology, adapted from Adeel et al. [1].

## 2.2. Data Collection and Analysis

In this section, we describe the methods used to test the methodology using real-world data. Using input from the workshops, we developed criteria (Table 1) to select major and case study flood events between 2013 and 2017 in Canada, Mexico, and the United States. We then collected quantitative economic cost data for these events to analyze data availability and accessibility. Finally, we selected three case studies to investigate the robustness of the methodology in different geographical and socio-economic settings in the three countries.

**Table 1.** Flood event selection criteria in this study.

| Major Flood Events (2013–2017) | |
|---|---|
| Required | It caused significant economic damages (e.g., cited as "most costly disaster) and significantly affected population or areas (e.g., affected by multiple municipalities or ≥1000 people affected). |
| Preferred | The government declared a national or provincial appeal (e.g., a declaration of a state of emergency). |
| **Case Study Flood Events (2013–2017)** | |
| Required | Prior reports and publications were available that allow the research team working on each case study to determine the inter-linkages between various events. Moreover, the economic impacts for the event(s) were well recorded and accessible. |
| Preferred | Each selected case-study event affected at least one Indigenous community or in both urban and rural communities or crossed state/province borders. |

We selected the following three case studies based on their goodness of fit with the aforementioned criteria: Fort McMurray (Canada) wildfire and flooding in May 2016, La Montaña region (Mexico) landslide and flooding in September 2013, and Louisiana and Texas (United States) flooding in March and August 2016. Although the subjectivity of the selection may bias the results, the team chose to examine data availability and accessibility in well-documented events; less significant events may have more data issues. See Supplementary Table S1 for a brief description of each case.

### 2.2.1. Data Collection

For both the major flood events and case studies, the final data collection scale was the secondary administrative level, which corresponds to the municipal level in Mexico, census division level in Canada, and county level in the United States (U.S.). (As described below, some data were collected at different geographic levels and then re-scaled to the secondary administrative level.) For each major flood event, we defined sub-flood events equal to each affected secondary administrative level. (For example, the 2013 Colorado flood (USA) impacted 19 counties, each defined in our data as a unique sub-flood event (19 sub-flood events).) The data collected in this study are secondary data that were previously estimated by another person or entity. Table 2 shows the complete list of sources used for data collection in each country.

**Table 2.** Data source for the major flood events and case study across Canada, Mexico, and the United States.

| Major Flood-Event Analysis | |
|---|---|
| County | Data source |
| Canada | Catastrophe Indices and Quantification Inc. (2013–2017) |
| Mexico | CENAPRED Book Series «Impacto Socioeconómico de los principales Desastres ocurridos en la República Mexicana» (2013–2017) |

**Table 2.** *Cont.*

| Major Flood-Event Analysis | |
| --- | --- |
| County | Data source |
| The United States | FEMA Disaster Declarations (2013–2017)<br>FEMA National Flood Insurance Program (2013–2017)<br>FEMA Individual Assistance Program (Homeowners and Renters Assistance) (2013–2017)<br>FEMA Public Assistance (2013–2017)<br>National Oceanic and Atmospheric Administration Storm Events (2013–2017)<br>U.S. Small Business Administration Disaster Assistance (Home and Business Loans) (2013–2017)<br>United States Department of Agriculture Risk Management Agency (2013–2017) |
| **Additional Data Sources Case Study Analysis** | |
| Canada<br>Mexico<br><br>The United States (See Section 3.3.3 for a discussion of how federal datasets are additionally used in the U.S. case study) | Wildfire: The Rapid Impact Assessment of Fort McMurray Wildfire (2017)<br>Toscana and Villaseñor [20]<br>U.S. Housing and Urban Development: Community Development Block Grant Program disaster fund allocations. Louisiana Office of Community Development, Disaster Recovery Unit (2021)<br>Private Insurance Marketplace damage and loss data (A. Smith, personal communication, 23 December 2020)<br>American Red Cross: costs for disaster services and supplies (2017) |
| **Auxiliary Data** | |
| Canada (Census data are utilized to change the geographic scale of the Canadian data (see supplementary materials Table S2)) | The 2016 Census Program, Statistics Canada |

Major flood event data collection:

In Canada, flood cost data were obtained from the Catastrophe Indices and Quantification Inc. (CatIQ). CatIQ is a subsidiary of the Zurich-based Perils AG and provides high-quality flood cost data on Canadian natural catastrophes. CatIQ datasets included flood insured losses by province and line of business (personal, commercial, and auto), divided into physical and non-physical damages. It should be noted that we reviewed the data available in the Canadian Disaster Database (CDD) and decided not to use it (see Section 4.1 for a detailed explanation). In Mexico, flood cost data from 2013–2017 were obtained from the series of books titled "Socio-economic impact of major disasters that occurred in the Mexican Republic," published annually by Centro Nacional de Prevención de Desastres (CENAPRED [21–24]). The Sub-directorate of Economic and Social Studies of CENAPRED compiled, organized, synthesized, and analyzed the data collected for the different government agencies.

The U.S. data sources included a combination of raw data collected from several different federal government-generated, open-access datasets. Major flood events were initially defined using the National Centers for Environmental Information's "Billion-Dollar Weather and Climate Disasters" [25]. Data collection included a desktop search of federal agencies and flood disaster programs and an examination of key U.S. flood-event damage data links and metadata that were available and accessible for 2013 to 2017. Additional datasets were identified and obtained from a search of the U.S. Government's open data website, Data.gov. Each data source provided a different type of information. In some cases, a single dataset, such as the United States Department of Agriculture's (USDA) Risk Management Agency (RMA) dataset, provided only one or two data elements that corresponded to a flood-event attribute or damage and loss indicator in the database. Other federal data sources, such as the Federal Emergency Management Agency's (FEMA) Public Assistance (PA) program, provided data for multiple damage and loss indicators.

Case study data collection:

Canada

To tabulate the costs of the 2016 Fort McMurray wildfire and flooding, a dataset from CatIQ (also used in the major flood events data) and an impact assessment led by MacEwan University (unique to the case study) served as sources of loss and damage data. (The 2016 Fort McMurray flood was one of the sub-flood events of the July 2016 Prairies Long Weekend Severe Storms. Thus, the Canadian case study flood data source (CatIQ) was the same data source that was used in the major flood-event analysis.) In the Rapid Impact Assessment of Fort McMurray Wildfire [26], led by a research team based at MacEwan University (Canada), preliminary estimates of the costs associated with the wildfire were tabulated. The researchers derived the estimates from many sources, including published data from Statistics Canada, Government of Alberta, and the Regional Municipality of Wood Buffalo; municipal property data and interviews with municipal and provincial officials; estimates based on existing literature; and statements reported in the media. Impacts were delineated into three categories: immediate direct impacts, long-term direct impacts, and indirect impacts.

Mexico

Due to the pandemic confinement decree by Mexican government authorities in 2020, bibliographic research was carried out to locate additional damage and loss data not analyzed by CENAPRED [21]. Obtained during field work in 2018, a study published by Toscana and Villaseñor [20] evaluated the performance of civil protection authorities and contained raw data for housing, rural schools, health units, and community bridges. Thus, these data were also included in the case study. In addition, CENAPRED data from 2013 [21] were used to populate the emergency response sector category.

The United States

The data collection involved a desktop review of federal, state, and municipal government disaster programs, Tribal government, Inter-Tribal organizations, and non-governmental organization websites, and published peer-reviewed literature. The process included the compilation of publicly available government documents, annual reports, and state-reported expenditure data to identify additional state and county damages and losses attributed to the 2016 flooding events. The National Oceanic and Atmospheric Administration (NOAA) provided private property insurance marketplace data for damages and losses in Louisiana and Texas. Additional costs reported in the open-access federal disaster datasets, including FEMA Individual Assistance (IA), FEMA PA, FEMA National Flood Insurance Program (NFIP), NOAA, Small Business Administration (SBA), and USDA RMA programs that were not included in the major flood-event analysis for the Louisiana and Texas flooding in March and August 2016, were used in this case study.

To compare damages from different countries and years, costs were adjusted to a common year and a single currency (real 2020 USD). To do so, economic costs of a flood event were first converted from local currency to US dollars using the exchange rate for the year of the event and then converted to 2020 USD using the Consumer Price Index, provided by the Bureau of Labor Statistics (available online at: https://www.bls.gov/cpi/ (accessed on 15 March 2021)). See supplementary Table S2 for details of data processing and analysis for specific countries.

## 3. Results

In this section, we briefly summarize major flood events that occurred in the three countries between 2013 and 2017. Then, we detail the results of the flood cost data availability and accessibility. Last, we present our three case study analyses.

### 3.1. Summary of Flood Events across Canada, Mexico, and the United States from 2013 to 2017

In total, 22 major flood events were collected from 2013 to 2017 in three countries (Table 3), including eight events in Canada, seven in Mexico, and seven in the United States. The eight Canadian flood events caused widespread damages across nine provinces, affecting 221 census divisions in Canada (Figure 2). Different types of storm systems or

rapid snowmelt caused these floods. The seven Mexico flood events caused immense damages and casualties across six states, affecting 320 municipalities (Figure 2). Torrential rain brought by hurricanes caused massive flooding, and these events were more frequent in the Pacific states, such as Baja California Sur, Guerrero, Chiapas, and Oaxaca. In the United States, seven flood events caused significant damages across six states, affecting 205 counties (Figure 2). Several moisture-laden systems brought abundant precipitation and flooding. In addition, the provinces of Saskatchewan, Manitoba, and Ontario, Canada; the state of Oaxaca, Mexico; and the state of Louisiana, United States, were each affected by multiple flooding events during the sample period.

The total damage from these 22 events was estimated at approximately USD 17 billion (real 2020 USD, same as below), averaging USD 3.40 billion per year in losses. (Given data gaps, this is likely a conservative estimate of total losses. See Section 3.2 for more information on data availability.) In Canada, total cost was estimated at USD 3.43 billion for the five-year period. In particular, total cost during the June 2013 Southern Alberta flood event was estimated at USD 1.68 billion (Table 3). In Mexico, total cost was estimated at USD 5.48 billion, and 149 people died from 2013 to 2017. The combined impacts of hurricanes Ingrid and Manuel, in September 2013, significantly affected the states of Guerrero and Nuevo León, killing 98 people and causing USD 3.08 billion in damages (Table 3). In the United States, total cost for the period was estimated at USD 8.1 billion. The seven U.S. flooding events killed 50 people. Around USD 5.18 billion in damages was estimated from the August 2016 Louisiana flood event (Table 3).

**Table 3.** Summary of flood events across Canada, Mexico, and the United States, 2013–2017 (unit: $ billions USD, real 2020 USD).

| Event No. | Date | Flood Events | Country | Flood Event Brief Description | Total Costs of Flood Events |
|---|---|---|---|---|---|
| 1 | May 2013 | Chiapas flooding | Mexico | Hurricane Barbara | USD 0.14 |
| 2 | June 2013 | Southern Alberta flooding | Canada | Persistent rain due to stationary system | USD 1.68 |
| 3 | July 2013 | Toronto flooding | Canada | Thunderstorm/flash flooding | USD 0.93 |
| 4 | September 2013 | Colorado flooding | United States | Flash flooding/debris flow | USD 0.67 |
| 5 | September 2013 | Guerrero flooding | Mexico | Hurricane Manuel | USD 2.87 |
| 6 | September 2013 | Nuevo León flooding | Mexico | Hurricane Ingrid | USD 0.21 |
| 7 | June 2014 | Southern Saskatchewan and Manitoba flooding | Canada | Persistent rain due to stationary system | USD 0.12 |
| 8 | August 2014 | Michigan and Northeast flooding | United States | Flash flooding/heavy rain | USD 0. 19 |
| 9 | September 2014 | Baja California Sur flooding | Mexico | Hurricane Odile | USD 1.82 |
| 10 | October 2015 | South Carolina and east coast flooding | United States | Flash flooding/heavy rain | USD 0.52 |
| 11 | March 2016 | Texas and Louisiana flooding | United States | Flash flooding/heavy rain | USD 0.66 |
| 12 | April 2016 | Houston flooding | United States | Flash flooding/heavy rain | USD 0.67 |
| 13 | June 2016 | Prairies and Northern Ontario flooding | Canada | Heavy rain | USD 0.03 |
| 14 | July 2016 | Prairie long weekend severe storms | Canada | Heavy rain | USD 0.37 |
| 15 | August 2016 | Louisiana flooding | United States | Flash flooding/heavy rain | USD 5.18 |
| 16 | August 2016 | Puebla flooding | Mexico | Tropical storm Earl | USD 0.19 |
| 17 | September 2016 | Windsor and Tecumseh Ontario flooding | Canada | Persistent rain due to stationary system | USD 0.13 |
| 18 | October 2016 | Nova Scotia, Prince Edward Island, and Newfoundland flooding | Canada | Hurricane Matthew | USD 0.08 |

**Table 3.** *Cont.*

| Event No. | Date | Flood Events | Country | Flood Event Brief Description | Total Costs of Flood Events |
|---|---|---|---|---|---|
| 19 | February 2017 | California flooding | United States | Severe winter storms, flooding, mudslides, and potential failure of the Emergency Spillway at Oroville Lake | USD 0.21 |
| 20 | May 2017 | Ontario and Quebec Spring flooding | Canada | Melting snow and ice | USD 0.09 |
| 21 | May 2017 | Oaxaca flooding | Mexico | Tropical storm Beatriz | USD 0.05 |
| 22 | June 2017 | Oaxaca flooding | Mexico | Tropical storm Calvin | USD 0.20 |
| | | | | | USD 17.01 |

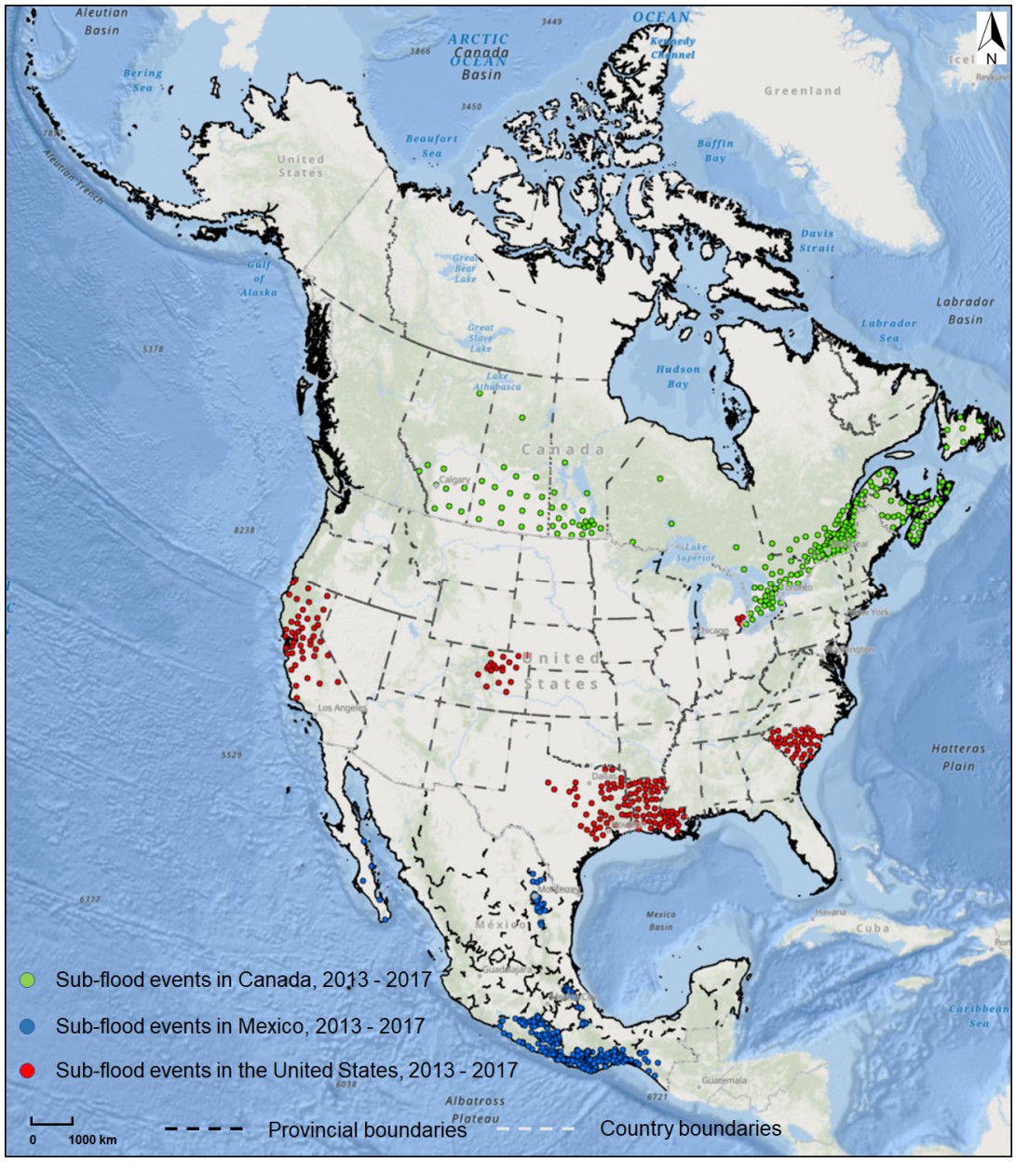

**Figure 2.** Spatial distribution of sub-flood events across Canada, Mexico, and the United States, 2013–2017.

*3.2. Data Availability for Flood Events across Canada, Mexico, and the United States from 2013 to 2017*

Applying the methodology, we found that Mexican damage data were the most complete among the three countries (Tables 4–6). For the seven flood events in Mexico, data were available for 61 to 93 of the 105 direct, indirect, and *loss and additional cost* indicators. Data were available for only 61 flood-damage indicators associated with the 2014 Baja California Sur flood—the fewest indicators of any Mexican flood incident. Among the seven flood events, the May and June 2017 Oaxaca flood events had the most complete data, with 93 flood damage indicators. For the United States, 14 indicators were available for the March 2016 Louisiana and Texas flood and August 2016 Louisiana flood—the most complete data availability among seven flood events. Limited flood cost data were collected by Canada. Data were available for only five damage indicators across the eight Canadian flood events. Moreover, small business interruption damages estimated from the CatIQ datasets only partially matched the credit damage indicator (e.g., indirect effect category, Table 5). Most of the Canadian records lacked data for the methodology damage indicators.

**Table 4.** Flood direct damage data availability and index coverage across Canada, Mexico, and the United States, 2013–2017 (Black color = data available; white color = no data).

| Direct Damage Categories | Direct Damage Indicators | Data Collection | | |
|---|---|---|---|---|
| | | Canada | Mexico | The United States |
| House | Household items | | ■ | ■ |
| | Dwelling | | ■ | ■ |
| | Cleaning | | ■ | |
| Education | Building | | ■ | |
| | Classroom | | ■ | |
| | Cleaning | | ■ | |
| Health | Death toll | | ■ | ■ |
| | Physical damage | | ■ | |
| | Medical equipment | | ■ | |
| Water and Sanitation | Storage tank | | ■ | |
| | Distribution network/treatment plant | | ■ | |
| | Rebuilding | | ■ | ■ |
| Cultural Resources | Place of worship | | ■ | ■ |
| | Recreation area | | ■ | ■ |
| | Sacred burial place | | ■ | |
| | Cultural artifact | | ■ | |
| | Museum collection | | ■ | |
| | Culturally relevant historic structure | | ■ | |
| | Damaged zone | | ■ | |

**Table 4.** *Cont.*

| Direct Damage Categories | Direct Damage Indicators | Data Collection | | |
|---|---|---|---|---|
| | | Canada | Mexico | The United States |
| Local Government/Community | Local infrastructure and services | | ■ | ■ |
| Transportation | Railroad | | ■ | |
| | Airport | | ■ | |
| | Port | | ■ | |
| | Road | | ■ | ■ |
| | Protection wall/dyke | | ■ | |
| | Restore the infrastructure | | ■ | |
| | Restore the services | | ■ | |
| Energy and Utilities | Power generation plant | | ■ | ■ |
| | Substation | | ■ | |
| | Transmission line and distribution grid | | ■ | |
| | Dispatch center | | ■ | |
| Technology and Communications | Service tower | | ■ | |
| | Communication infrastructure | | ■ | |
| Agriculture | Road or bridge | | ■ | |
| | Storage space | | ■ | |
| | Infrastructure used in farming | | ■ | ■ |
| | Infrastructure used in livestock | | ■ | |
| | Infrastructure used in poultry | | ■ | |
| | Infrastructure used in private forestry activity | | ■ | |
| Fisheries | Storage space | | ■ | |
| Manufacturing | Building and facility | | ■ | |
| | Machinery and equipment | | ■ | |
| | Inventory of goods | | ■ | |
| Commerce | Building and facility | ■ | ■ | ■ |
| | Machinery and equipment | | ■ | |
| | Inventory of goods | | ■ | |
| Tourism | Tourism area | | ■ | |
| | Property | | ■ | |
| Public Forest | Employee | | ■ | |
| | Road or bridge | | ■ | |
| | Infrastructure used in the park | | ■ | |
| Environment | Erosion and sedimentation | | ■ | |
| | Wildlife and aquatic species health | | ■ | |
| | Dispersal of nutrients and pollutants | | ■ | |
| | Local landscapes and habitats | | ■ | |

**Table 5.** Flood indirect effect data availability and index coverage across Canada, Mexico, and the United States, 2013–2017 * (Black color = data available; white color = no data; grey color = data available for a subset of the indicator).

| Indirect Effect Categories | Indirect Effect Indicators | Data Collection | | |
|---|---|---|---|---|
| | | Canada | Mexico | The United States |
| House | House rental | | ■ | |
| Education | Missing workdays due to school closure | | | |
| Health | Patient | | | |
| | Workdays lost. Missing workdays due to psychological impacts, stress, and anxiety | | | |
| Local Government/Community | Workdays lost (unemployment increases) | | | |
| Transportation | Loss of revenue at ports | | ■ | |
| Energy and Utilities | Spills damage | | ■ | |
| Technology and Communications | Revenue (manufacturing) | | ■ | |
| | Revenue (commerce) | | ■ | |
| Public Infrastructure | Non-market value of public space | | ■ | |
| Manufacturing | R&D impacts | | ■ | |
| | Loss of wages, including temporary jobs | | | |
| Commerce | Credit. Decreased credit scores and bond downgrades for businesses | ▨ | | |
| Tourism | Loss of wages | | | |
| Public Forest | Workday lost | | ■ | |

* It should be noted that data for eight indirect effect indicators were available in all Mexican flood events, and they were all identified as no damage (zero is identified as no damage).

**Table 6.** Flood *losses and additional cost* data availability and index coverage across Canada, Mexico, and the United States, 2013–2017 (Black color = data available; white color = no data).

| Losses and Additional Costs | Losses and Additional Cost Indicators | Data Collection | | |
|---|---|---|---|---|
| | | Canada | Mexico | The United States |
| House | Temporary accommodation | ■ | ■ | ■ |
| | Relocation | | ■ | |
| Education | Temporary classroom | | ■ | |
| | Reset service | | ■ | |
| Health | Post-disaster epidemic | | ■ | |
| | Hospital-related costs | | ■ | |
| | Structure-related costs | | ■ | |
| Water and Sanitation | Temporary water needs | | ■ | |

**Table 6.** *Cont.*

| Losses and Additional Costs | Losses and Additional Cost Indicators | Data Collection | | |
|---|---|---|---|---|
| | | Canada | Mexico | The United States |
| Cultural Resources | Revenue (cultural resources). Loss of revenue to religious/cultural organizations | | ■ | |
| | Recreation. Loss of recreation services (non-market values) | | ■ | |
| Local Government/Community | Revenue | | | |
| | Loans and bonds | | | |
| | GDP | | | |
| Transportation | Cost for transporting freight | | ■ | |
| | Loss of tolls | | | |
| | Cost for passengers | | | |
| | Additional costs for crews | | ■ | |
| Energy and Utilities | Revenue | | ■ | |
| | Rehabilitation/reconstruction | | ■ | |
| Public Infrastructure | Cleaning | | ■ | ■ |
| | Rescheduling public events' costs | | ■ | |
| Agriculture | Market value of crop | | ■ | ■ |
| | Income | | ■ | |
| | Market value of livestock | | ■ | |
| | Market value of poultry | | ■ | |
| | Market value of private forest product | | ■ | |
| Fisheries | Market value of fish | | ■ | |
| | Market value of crustaceans | | ■ | |
| | Income | | ■ | |
| Tourism | Service flow | | ■ | |
| Public Forest | Market value | | ■ | |
| Emergency Response | Transporting the wounded or other emergency evacuations | | ■ | |
| | Equipment | | ■ | |
| | Temporary shelters | | ■ | |
| | Search for people | | ■ | |

We found large variations in the type of data available across countries and damage categories during our 2013 to 2017 scope. See Section 4.1 for a discussion of data availability and collection challenges. The most complete information was available for direct flood damage indicators. Notably, data availability for direct damages was 100% (55 out of 55 indicators) for the May and June 2017 Oaxaca flood events. In contrast, only nine out of fifteen indirect effect indicators had data available for all flood events across the three countries (Table 5). Although data for eight indirect effect indicators were available in all Mexican flood events, they were reported as no damage (e.g., zero is identified as no damage). Combined with the other seven indirect effect indicators identified as no data (−999 is identified as no data), there were no recorded indirect flood-related economic costs identified in all Mexico flood events between 2013 and 2017. Data were available for the credit damage indicator across all eight of the Canadian flood events. Moreover, based on several federal government flood databases, none of the seven flood events in the United States contained any data for indirect effects. However, indirect effect data were found from the state-level data for the August 2016 Louisiana flood (Section 3.3.3). The records from Canada and the United States populated fewer than 10% of the flood *loss and additional cost* indicators (Table 6). Among 35 *loss and additional cost* indicators, only costs of the provision of temporary accommodation data were collected in Canadian flood events. A maximum of three *loss and additional cost* indicators were available and collected in the United States flood events. Data were available for most of the *loss and additional cost* indicators for

Mexican flood events; however, there were gaps for local government/community and transportation data in Mexico (Table 6).

Flood damages in the housing sector were the most comprehensive of data available in the methodology categories across the three countries between 2013 and 2017. All flood events had data on the damages of household items, dwellings/properties, and costs of the provision of temporary accommodation for persons whose homes were destroyed or had to be abandoned. Data for the cost of total or partial destruction of commercial buildings and facilities also were available for all three countries.

### 3.3. Case Study Analysis

Compared with our major flood-event analysis, our case study analysis revealed additional flood cost data across the three countries (e.g., flood cost data from local communities/governments and other research studies). Although Mexico showed the most complete data availability across the three countries in the major flood-event analysis, there were additional flood cost data from local studies. Similar to our major flood-event analysis, case study analyses also indicated that very limited flood-related indirect effects and *losses and additional costs* were documented across the three countries. In particular, few indirect effect indicators were available across the three case studies. Obtaining fine-scale data was challenging. Many data were highly aggregated, only indicating a total cost from one sector and not specific indicators. Below, we detail our three case study results.

### 3.3.1. The 2016 Fort McMurray Wildfire and Flooding, Province of Alberta, Canada

The 2016 flood was contained to the city of Fort McMurray, while the area affected by the wildfire was far greater (Figure 3). The total cost of the cascading event was estimated to be nearly USD 6.5 billion (Table 7, real 2020 USD). The wildfire proved to be far more costly than the flood. At approximately USD 1.7 billion (real 2020 USD), the costliest indicator was the market value of forest products (wildfire), amounting to 27.2% of the total cost of the cascading event. Other costly indicators included dwelling damage (wildfire and flood), revenue loss for the energy sector (wildfire), and commerce building damage (wildfire and flood), amounting to 23.5%, 21.3%, and 15.4% of the total cost of the cascading event, respectively. The least costly indicator was psychological impacts (wildfire), amounting to a very small fraction (0.3%) of the total cost of the cascading event. *Losses and additional costs* comprised 53.8% of the total cost of the cascading event. Direct damages amounted to 43.4%. Insured and uninsured costs of the cascading event were split somewhat evenly. Estimated insured losses were 46% of the total cost of the cascading event, with uninsured costs comprising a slight majority at 54%.

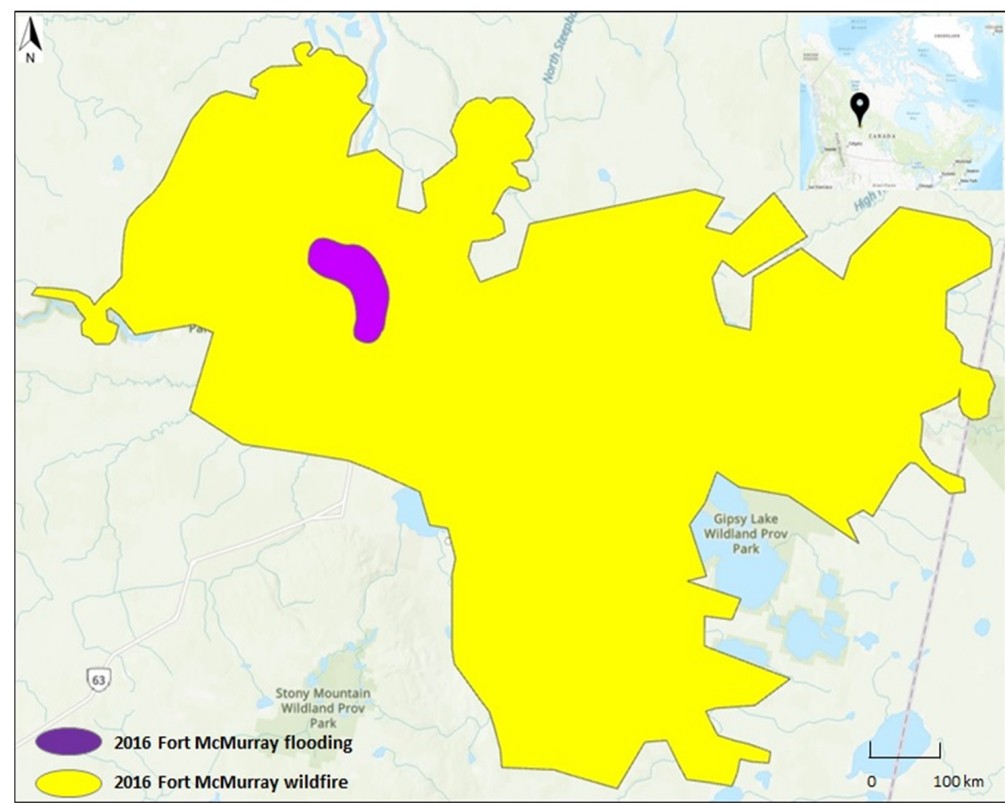

**Figure 3.** Geographical overview of the 2016 Fort McMurray wildfire and flooding, Canada, adapted from CatIQ [27].

**Table 7.** Estimated damages and losses caused by the 2016 Fort McMurray wildfire and flooding, Canada (unit: real 2020 USD).

| Indicator | Wildfire | Flooding | Total Damage |
|---|---|---|---|
| Household item (direct damage) | USD 87.8 million | USD 5.3 million | USD 93.1 million |
| Dwelling (direct damage) | USD 1514 million | USD 6.4 million | USD 1520.4 million |
| Commerce building and facility (direct damage) | USD 994 million | USD 1 million | USD 995 million |
| Commerce credit (indirect effect) | USD 154 million | USD 0.01 million | USD 154 million |
| Temporary accommodation (*loss and additional cost*) | USD 194.2 million | USD 6000 | USD 194.2 million |
| Distribution network treatment plant (direct damage) | USD 51.8 million | Not available (N/A) | USD 51.8 million |
| Erosion and sedimentation (direct damage) | USD 148 million | N/A | USD 148 million |
| Psychological impacts, stress, and anxiety (indirect effect) | USD 21.8 million | N/A | USD 21.8 million |
| Loss of tax revenue for local governments (*loss and additional cost*) | USD 150.4 million | N/A | USD 150.4 million |
| Loss of revenue for energy and utilities (*loss and additional cost*) | USD 1375 million | N/A | USD 1375 million |
| Market value of public forest product (*loss and additional cost*) | USD 1761 million | N/A | USD 1761 million |
| **Total damage** | **USD 6452 million** | **USD 12.71 million** | **USD 6464.7 million** |

3.3.2. The 2013 La Montaña Region Landslide and Flooding, State of Guerrero, Mexico

The combined impacts of the September 2013 Hurricanes Ingrid and Manuel caused significant economic damage in the La Montaña region. Applying the methodology, data were available for six sectors, including housing, health, education, hydraulic infrastructure, urban infrastructure, and emergency care (Figure 4). The total cost of the flood event in this area was approximately USD 83 million (real 2020 USD, same as below). The total direct damages for the six sectors were USD 77.52 million, while the additional losses amounted to USD 5.51 million (Figure 4). Urban infrastructure and the education sector had the highest economic costs, representing 35.8% and 53.9% of the total economic impact, respectively.

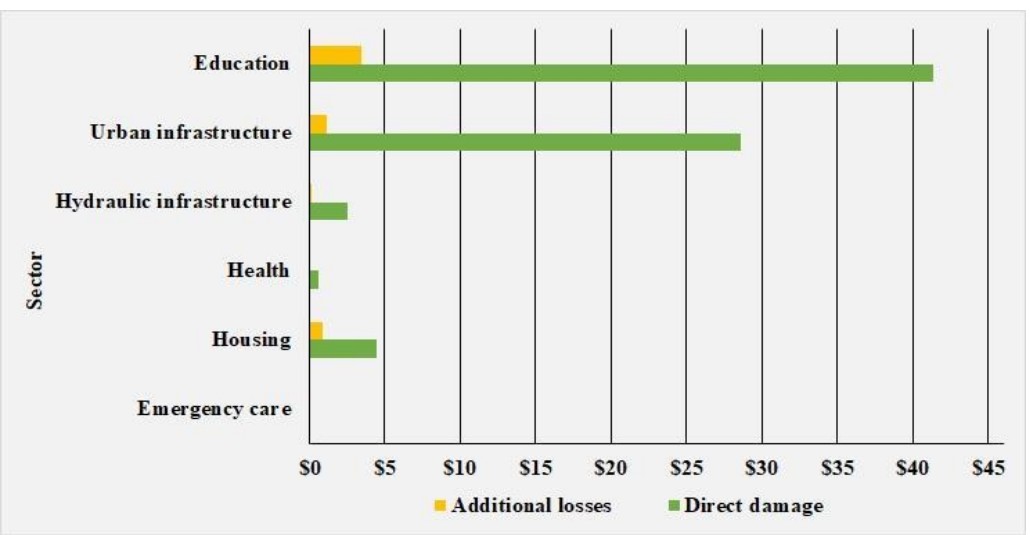

**Figure 4.** Estimated damages and losses in the region of La Montaña, State of Guerrero, Mexico, caused by Hurricanes Ingrid and Manuel in 2013 (unit: $ millions USD, real 2020 USD).

The socioeconomic impact of Hurricanes Manuel and Ingrid in the state of Guerrero was evaluated by CENAPRED. However, in the mountainous regions of La Montaña (Figure 5), several landslides occurred on rural roads and bridges, which limited full access to this area. Thus, among the sectors indicated in Figure 4, some rural roads and bridges and small farming areas were not accounted for due to this inaccessibility. In 2018, Toscana and Villaseñor [26] conducted a study of government actions in response to the events that occurred in the area in 2013. While not the central objective of their analysis, during the fieldwork they nonetheless found impacts to 2988 homes, 540 rural schools, 35 health centers, and 135 federal highways. These raw data are considered additional to those compiled by CENAPRED, with the exception of those associated with emergency care.

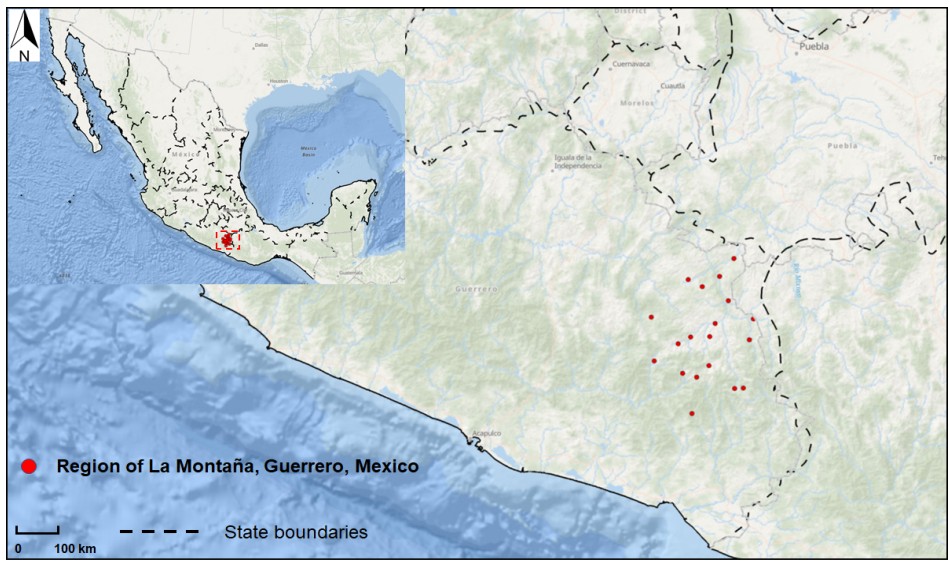

**Figure 5.** Geographical overview of the region of La Montaña, State of Guerrero, Mexico.

### 3.3.3. The 2016 Louisiana and Texas Flooding, the United States

The 2016 flooding extended across coastal and inland areas of Louisiana and eastern Texas, impacting more than 80 counties in both states (Figure 6). Under the methodology, the direct damages and additional losses for the March 2016 flooding in Louisiana and Texas and the August flooding in Louisiana totaled nearly USD 662 million (real 2020 USD, same as below) and more than USD 5.18 billion, respectively. Through the case study analysis, we collected and included an additional USD 128.87 million in flood damages associated with the March 2016 flooding, including USD 17.41 million in Texas and USD 111.46 million in Louisiana and nearly USD 721 million associated with the August flooding in Louisiana. As noted in Table 2 in Section 2.2.1, sources for these additional data include costs reported in the federal assistance data (FEMA IA, FEMA PA, NFIP, and SBA), state of Louisiana disaster aid and recovery expenditures, and other governmental and non-governmental entities. Following the methodology, these additional impacts were not included in the major flood-event analysis (Section 3.2) due to data issues, including a high level of data aggregation, costs related to mitigation (which are excluded from the methodology), and difficulty in defining and attributing costs to specific methodology indicators. (For example, the methodology includes only economic costs at the county level, but the FEMA PA data include additional damages and disaster management costs associated with state government sectors and services. See also Section 4.1 for additional discussion on data availability across the methodology and case study.)

The combined state-level damages and costs, totaling nearly USD 850 million for the March and August 2016 floods in Louisiana and Texas, are itemized and referenced by the type of damage and loss, sector category, and methodology indicator category (as applicable) in Table 8. Of these losses, 84.8% were impacts associated with the August Louisiana flooding, while 13.1% were associated with the March Louisiana flooding and 2.1% with the Texas flooding. The highest costs were associated with the emergency assistance sector, accounting for an estimated 55.8% of the total additional damages and losses, followed by the social services sector, making up an estimated 30.5% of the additional costs (Table 8).

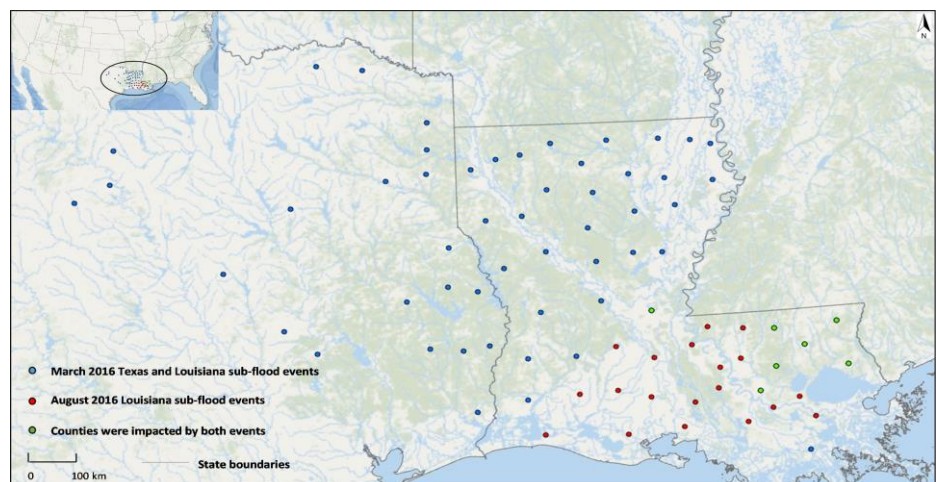

**Figure 6.** Geographical overview of the 2016 Louisiana and Texas flooding, the United States.

**Table 8.** Additional state and county level costs caused by the 2016 Louisiana and Texas flooding, the United States (Unit: $ millions USD, real 2020 USD). (All costs listed in Table 8 represent additional costs uncovered in the case study and are not included in the flood events' results based on the methodology.).

| Type of Damage and Loss | Texas March Flooding | Louisiana March Flooding | Louisiana August Flooding | Total Damage |
|---|---|---|---|---|
| **Social sectors** | | | | |
| Housing damages for other counties not included as part of the federal disaster declaration [1] (direct damage) | USD 0.13 | USD 1.93 | USD 0 | USD 2.06 |
| Housing damages for increased cost of compliance [1] (direct damage) | USD 0.35 | USD 1 | USD 19.56 | USD 20.91 |
| Other needs assistance for individuals [2] (*loss and additional cost*) | USD 3.05 | USD 27.3 | USD 176.77 | USD 207.12 |
| State debris removal and clean-up [3] (*loss and additional cost*) | USD 0.38 | USD 0.91 | USD 7.74 | USD 9.03 |
| Statewide management costs [3] (*loss and additional cost*) | USD 0.66 | USD 3.37 | USD 16.2 | USD 20.23 |
| **Infrastructure sectors** | | | | |
| State utility-related impacts [3] (direct damage) | USD 0.97 | USD 1.07 | USD 6.34 | USD 8.38 |
| State buildings, facilities, and equipment [3] (direct damage) | USD 0.09 | USD 4.97 | USD 13.11 | USD 18.17 |
| State impacts to parks and recreational facilities [3] (direct damage) | USD 0.03 | USD 0.95 | USD 0.1 | USD 1.08 |
| State impacts to roads and bridges [3] (direct damage) | USD 0.15 | USD 3.05 | USD 6.22 | USD 9.42 |
| State water infrastructure impacts [3] (direct damage) | USD 0 | USD 1.11 | USD 0.07 | USD 1.18 |

**Table 8.** *Cont.*

| Type of Damage and Loss | Texas March Flooding | Louisiana March Flooding | Louisiana August Flooding | Total Damage |
|---|---|---|---|---|
| **Economic sectors** | | | | |
| Business economic injury (federal loans) [4] (indirect effect) | USD 0.4 | USD 15.35 | USD 41.07 | USD 56.82 |
| Louisiana farming sector impacts and recovery [5] (direct damage/loss and additional cost) | N/A | USD 10.64 | USD 10.64 | USD 21.28 |
| **Emergency assistance** | | | | |
| Emergency food, shelter, and relief items [6] (*loss and additional cost*) | | | USD 19.3 | USD 19.3 |
| Health and emotional support services [6] (indirect effect) | | | USD 1.72 | USD 1.72 |
| State and county emergency and protective response measures [3] (*loss and additional cost*) | USD 11.2 | USD 39.81 | USD 402 | USD 453.01 |
| **Total damage** | **USD 17.41** | **USD 111.46** | **USD 720.84** | **USD 849.71** |

[1] Housing-related costs were obtained from the NFIP data. [2] Other needs assistance costs were obtained from the FEMA IA data. [3] State-level costs were obtained from FEMA PA data. [4] Economic injury costs were obtained from SBA business disaster loan data. [5] Damages and losses for the farming sector that are attributed to the March and August flooding were estimated by the U.S. Housing and Urban Development: Community Development Block Grant Program disaster fund allocations [28] and aggregated by the State of Louisiana in annual expenditure reports. [6] Costs for disaster services and supplies, as reported by the American Red Cross [29].

In addition, the U.S. Department of Housing and Urban Development (HUD) awarded a total of USD 520 million (real 2020 USD) in recovery assistance to Louisiana and Texas for damages and losses resulting from the March and August 2016 floods [30]. The funding was provided through HUD's Community Development Block Grant—Disaster Recovery (CDBG-DR) Program and intended to target unmet needs and provide housing assistance to the most impacted communities [30]. The CDBG disaster grant program does not duplicate the damage and loss payouts of other federal disaster programs. The CDBG costs are dispersed to various state subprograms and projects, and some portion of these damages may be designated for mitigation. An estimated 85% of the state of Louisiana's shares of the CDBG funds were allocated to the infrastructure program for housing-related damages and losses (Figure 7, [28]).

More than 90% of U.S. residential flood insurance is provided through the NFIP, with less than 10% of remaining policies covered through the private insurance marketplace (A. Smith, personal communication, 23 December 2020). Figure 8 provides a breakdown of the private insurance costs for the March and August flooding. The private insured damages and losses associated with the August 2016 flooding in Louisiana totaled USD 990 million (Figure 8, [31]). Of these losses, approximately 15% were commercial, 10% were residential, and 75% were automotive. Although the total private insured losses associated with the March flood were less than those of the August event, the costs were still substantial. The Texas private insured losses to commercial, residential, and automotive assets totaled USD 105 million. Of these losses, approximately 24.8% was commercial, 15.2% was automotive, and 60% residential. The private insured losses in Louisiana for March 2016 totaled USD 110 million, of which 10% was commercial, 40% was residential, and 50% was automotive.

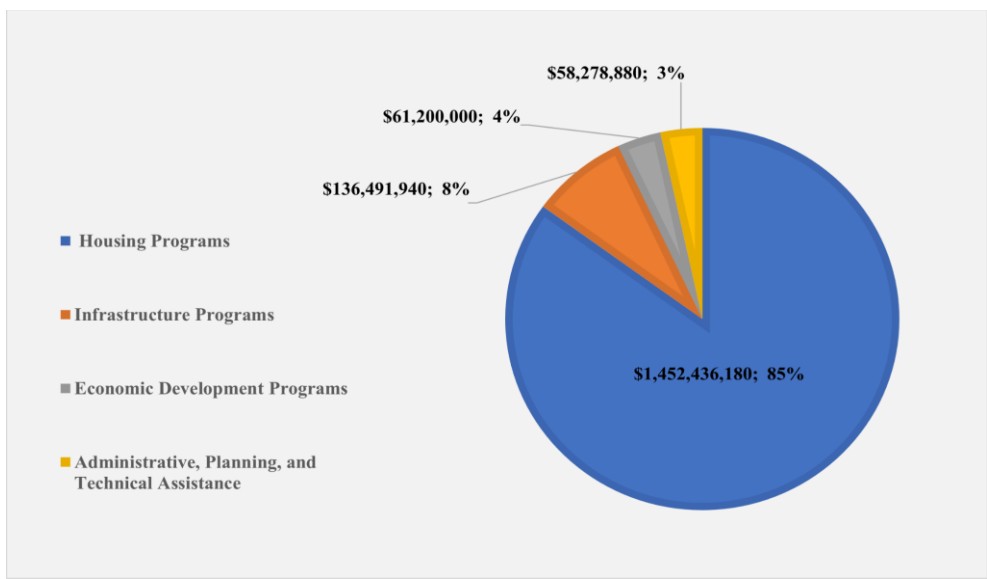

**Figure 7.** Distribution of U.S. Housing and Urban Development Grant assistance for the state of Louisiana by program category. Data from: [28] (unit: real 2020 USD).

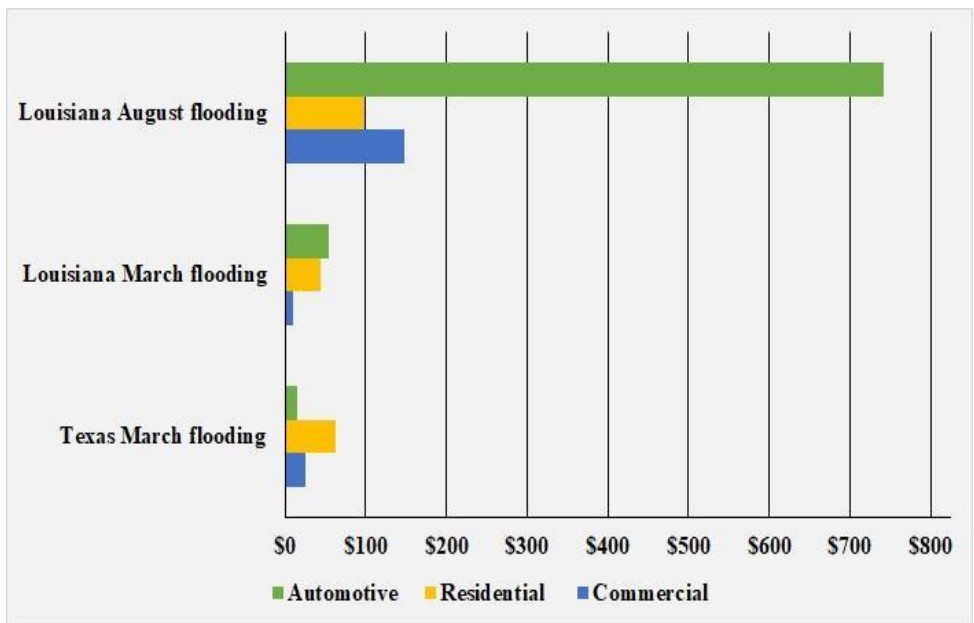

**Figure 8.** Private property marketplace insured damages and losses (unit: $ millions USD, real 2020 USD). Data from: [31].

## 4. Discussion

In this section, we first discuss flood cost data availability and accessibility by county and sector across the three countries. Next, we detail flood cost data collection challenges and research limitations and opportunities.

### 4.1. National Flood Cost Data Accessibility and Availability

Based on our test period (2013–2017), economic cost data for flooding in Mexico were comprehensive, closely matching the methodology. This result was expected, as the methodology was largely based on the adapted UN-ECLAC methodology used in Mexico, and it verified that Mexico had implemented that methodology effectively. The data on the socioeconomic impact of the floods in Mexico are compiled by CENAPRED, which is

the federal government agency in charge of this task. Data also are collected by different agencies that have a presence at the state and municipal level in the affected territory. Then, CENAPRED performs a thorough analysis of the data and events and prepares a report and recommendations to the corresponding authorities and decision makers. Data are usually classified in four main categories such as social sectors (e.g., housing, education, health, etc.), economic infrastructure (e.g., communications, transports, energy, etc.), productive sectors (e.g., tourism, fishing, agriculture, etc.), and emergency care. In some instances, data from the private insurance sector are also integrated into CENAPRED datasets.

In Mexico, flood economic damage data compiled by CENAPRED are considered public information that is used by, and benefits, many sectors. An example is the data benefit emergency response services for impacted communities. Coastal communities live off tourism and, by using these services, they can be aided in economic recovery after an event. Remote communities depend on agriculture, and if their lands are flooded and they cannot store seeds or harvest food, they need assistance from the government. The flood database also helps CENAPRED identify the most vulnerable communities, formulate efficient response strategies, and develop policies that can address challenges. Moreover, it is worth mentioning that flood cost data from Mexican flood events have partially documented damages from Indigenous communities. For example, the flood events of Chiapas-Guerrero (2013), Puebla (2016), and Oaxaca (2017) caused multiple local impacts in communities where the population was comprised of both Indigenous and non-Indigenous people, and in sectors such as the highway, housing, and agriculture. Through CENAPRED's leadership, substantial progress has been made on flood cost data collection and management in Mexico. The national disaster fund is managed by a centralized government agency, and a certain amount of money is allocated for disaster response and recovery. However, we found data gaps in some municipalities. Participation of key players in local communities is needed to strengthen flood economic cost data collection in Mexico.

Unlike Mexico, data collection in Canada and the United States is undertaken by multiple agencies that focus on different jurisdictions and scales of flood damage and loss, making data collection and impact assessment complicated and incomplete. In Canada, flood damage data can be collected at different scales by different agencies (e.g., federal, provincial, municipal governments or departments, and private sectors). Similar to Canada, multiple governments (e.g., federal, state, local, and tribal governments), non-governmental organizations, and private sectors can provide flood damage and cost data in the United States. Compared to Canada, different U.S. federal government-generated, open-access datasets are easily accessible. Moreover, some flood cost data cannot be shared with the public in Canada and the United States because of data privacy (e.g., data contains personally identifiable information) and data sensitivity. For example, some U.S. federal agencies (e.g., U.S. Army Corps of Engineers) may not provide disaggregated or consistent data for damages and losses to critical infrastructure (e.g., dams and levees) due to national security concerns.

Public flood cost data exist in Canada, such as the CDD and Provincial Disaster Assistance Program (PDAP). A common feature of these public flood cost data is that data are highly aggregated. This aggregation is the major reason that we excluded the CDD and PDAP datasets in our analysis. The CDD dataset did not indicate which sector claimed the loss and only provided an estimated total cost. Moreover, the CDD data sources include the federal Disaster Financial Assistance Arrangements (DFAA), provincial DFAA, provincial department payments, municipal costs, insurance payments, and non-governmental organization (NGO) payments. In fact, most flood cost data between 2013 and 2017 were collected from federal DFPP and insurance payments. Data from provincial department payments, municipal costs, and NGOs were marked as "Unknown" in the CDD dataset. We also collected the PDAP data from Saskatchewan and New Brunswick between 2013 and 2017, but these two PDAP datasets only provided an estimated total cost of sector damages. Moreover, many flood events were small scale between 2013 and 2017.

Although we did not use the CDD and PDAP datasets in our analysis, these data are useful to identify the total cost of flood and provide a sense of their scale.

In the U.S. major flood-event analysis, data sources included a combination of raw data collected from several different federal government-generated, open-access datasets (see Section 3.3.3 for details). The reporting of damages and losses by federal agencies for major floods may occur over several years and, according to FEMA, the updates to federal datasets are variable, with some occurring daily and others quarterly. While care must be taken in combining data to ensure that losses are not counted twice, most of the federal datasets, including those reported by the FEMA, are managed across multiple federal disaster programs to avoid the duplication of damages. However, based on our U.S. case study analysis, we found it was difficult to determine if some state-reported flood economic costs are duplicated in other sources. (For example, state government processes for reporting damages and losses are highly variable and states may be reporting allocations of damages already included in federal damage datasets. Some state-reported expenditures may be distributed to specific counties for damages and losses that are similarly reported in federal disaster datasets. Moreover, state agencies may combine costs and allocate funds for multiple events, and it is difficult to discern which damages are attributed to a particular flood event.) Therefore, we excluded the state-level flood economic costs data in the major flood-event analysis to avoid double counting.

We used the secondary data sources; thus, the existing datasets did not always contain measures that aligned with the categories and indicators we defined in the methodology. For the U.S. major flood-event analysis, we had to exclude many data due to mismatched flood damage definitions. For example, indirect effects that we defined in the methodology were sometimes coupled with direct damages in other datasets. Likewise, the damages and losses in existing datasets were defined and categorized differently and might not reflect the definitions of the indicators in the methodology. (For example, in the United States, the FEMA IA dataset aggregates multiple types of damages and losses under one category, whereas the methodology has specific indicators for each type of damage and loss.) Although some indicators defined in the methodology were similar to indicators defined in other datasets, in some cases, it was difficult to accurately link the damages and losses to the methodology indicators. Thus, our study provides a conservative lower bound on economic estimates of flood damages, in comparison with mainstream federal government-generated, open-access datasets such as NOAA (Table 9, [24]). Similarly, we provide conservative lower bound flood economic cost estimates for the Canadian flood events, because we exclude the CDD and PDAP datasets.

**Table 9.** Comparison between NOAA event estimated cost and estimated cost under the methodology (unit: $ millions USD, real 2020 USD).

| U.S. Flood Events (2013–2017) | NOAA Event Estimated Cost * | Estimated Cost under the Methodology |
|---|---|---|
| 2013, Colorado flooding | USD 1700 | USD 674 |
| 2014, Michigan and Northeast flooding | USD 1100 | USD 188 |
| 2015, South Carolina and east coast flooding | USD 2200 | USD 518 |
| 2016, Texas and Louisiana flooding | USD 2500 | USD 662 |
| 2016, Houston flooding | USD 2900 | USD 667 |
| 2016, Louisiana flooding | USD 11,000 | USD 5182 |
| 2017, California flooding | USD 1600 | USD 209 |

* Data from: [24].

### 4.2. Sectoral Flood Cost Data Accessibility and Availability

Flood economic damage data were unevenly collected by sectors in Canada, Mexico, and the United States between 2013 and 2017. Moreover, compared to flood cost data

sources in Mexico and the United States, flood economic cost sectoral coverage was very limited in Canada. On the other hand, we found multiple U.S. datasets that provided damage and loss data for the housing sector, including the FEMA NFIP, FEMA IA, and SBA, which provided disaster assistance for damages and losses associated with real estate and contents. In the housing sector, except for the 2017 California flooding, household item and dwelling damage data were collected from at least two data sources. (For example, both FEMA and SBA datasets offered dwelling damages for the 2013 Colorado flooding and household item damages for the 2016 Houston flooding, respectively.) Under the methodology, the true total indirect flood economic costs across the three countries were undercounted, as very limited indirect effect data were collected. Moreover, records from Canada and the United States populated few of the defined flood *losses and additional cost* categories. Consequently, there remain key data gaps in additional flood costs in Canada and the United States.

Understanding all flood risks is crucial for better preparing post-flood recovery strategies and fully assessing the cost–benefits of adaptation. Many studies have highlighted that underestimating indirect losses and additional effects misguides flood risk mitigation and adaptation, as flood indirect impacts may cause significant economic damages [32–34]. For example, Carrera et al. [33] estimated indirect costs of the Po River October 2000 flood event in Italy, and their results showed that the estimation of indirect costs ranged from EUR 3.3 to EUR 8.8 billion (real 2000 Euro). However, quantifying all flood risks is also challenging. A flood event can last days or weeks, and its impacts (e.g., physical and psychological health) can last months or years [35]. Some chronic physical and emotional conditions might appear in the months following a flood. Allocations and reporting of flood damages, especially indirect effects and *losses and additional costs*, may occur over years.

*4.3. Research Opportunities, Limitations, and Recommendations*

Detailed priorities for action to reduce disaster risk under the Sendai Framework include promoting data collection, analysis, management, use, and dissemination; systematically evaluating disaster losses and their economic impacts; and improving collaborations among scientific, technological, stakeholder, and policy communities for integrating science in disaster risk management [9]. In this study, however, we found substantial gaps in each of these areas. Especially, under the methodology, the most significant challenge in data collection was to tackle the systemic data vacuum on disaster damages and losses across the three countries, especially in Canada. Throughout the data collection process, obtaining granular data on disaster damages and losses at the municipal/county/census division level proved to be a challenge. In Canada, much of the existing flood cost data, even wildfire cost data (based on our case study analysis), were aggregated by province, and categories were generalized. We addressed this challenge through the population weighting adjustment to disaggregate the data, but a better solution is to collect data at this more spatially refined level. In Mexico, flood economic damage data were available at the municipal scale. However, missing data were across multiple municipalities. Mexico struggles with tools and capacities to collect and manage data at the municipal level. Technical assistance, particularly at the municipality level, in Mexico is needed because they are the ones reporting their damages and losses. Some municipalities do not have internet or laptops, making data collection and management difficult. There is an urgent need to undertake capacity building and training for data collection and management. In the United States, different federally operated, open-access flood cost datasets are easily accessible, while detailed state-, tribal-, and county-level damage data and county-level private property insured data were not easily accessible. Working with the insurance market provides an opportunity to combine efforts and have access across sectors, institutions, and academia. However, insurance availability and uptake for many communities is low and different across the three countries. Quantifying uninsured losses is also challenging.

Our study provides a foundation to bring consistency to data collection and management across Canada, Mexico, and the United States. Our study provides (1) insights

to compare differences among how the three countries collect post-flood cost data; and (2) contributions to better understand the strengths and weaknesses of data availability and access across the three countries. However, there are several limitations of this flood cost analysis. First, the 22 flood events were derived from major flood events across the three countries, which excluded economic impacts from small but perhaps more frequent flood events. Second, we used secondary data in this study. Secondary data that have been widely used in many fields [36,37] can be economical, efficient, and time-saving for addressing research questions and can be used to address new research topics that are not associated with the originally published analysis of the data [38]. However, the user of secondary datasets cannot control the scales and methods of measurement. In this sense, secondary data analysis is not always suitable for other research questions. We excluded some flood cost data due to a high level of data aggregation and different definitions for flood damages. Thus, our flood economic cost analysis is conservative due to under-recorded or unrecorded data. In particular, we stress that we are not aiming to provide a better estimate of aggregate total flood costs. One of the biggest contributions of this study is to collect and carefully categorize spatial, temporal, and outcome-specific localized flood economic costs. In the process, we provide a better understanding and more detailed insights into flood economic cost data gaps for the three countries. Third, we acknowledge that additional data sources are available in the three countries, including uninsured and insured flood damages at the provinces/states and municipalities. Based on our case study analysis, especially the U.S. case study, we found additional flood cost data from the state government. We also acknowledge that an extended time window (e.g., a ten-year testing window), although beyond the scope of the CEC project, would better assess the applicability and robustness of this proposed methodology [1].

Although the methodology defined 105 damage indicators that cover direct, indirect, and additional damages, we found that flood economic damage assessments rarely capture the indirect damage across the three countries. Thus, prioritizing the dozens of missing flood damage indicators will aid in flood response and resilience building at local levels. Next steps in the application of the methodology could be achieved through local-level pilot tests across the three countries, such as directly working with different users and communities, to develop criteria for priority setting for implementing flood damage indicators in practice. In addition, methodology data could be coupled with additional geospatial community resilience, flood hazard, and response data to better inform emergency management, mitigation activities, and flood planning that are critical to combat these disastrous events now and into the future.

Our results revealed the need for an open-access, centralized flood cost data center where data should be standardized and made interoperable. A centralized flood cost data center could have further benefits for (1) the insurance industries, to improve the assessment of financial risk related to flood coverages; and (2) for citizens, to increase their understanding of territorial risk, thus increasing their confidence in undertaking mitigation actions. However, to capture flood-related economic damages requires a broad array of public and private data. Coupling the data sources within an integrated framework also has its challenges. Currently, Canada, Mexico, and the United States lack a common flood economic damage data center or hub. There is an urgent need to encourage regional, and even international, cooperation and coordination to develop hazard methodology and data standards. However, jurisdictional complexities make it difficult to gather and manage flood information across the countries' borders, even across provincial or state borders. Compared to Canada and the United States, the structure in Mexico is more centralized, making for easier flood data collection and management. Standardized scientific and mapping information to better communicate flood hazard and risk is a challenge, given the many jurisdictions involved and the complexity of coordinating across many relevant stakeholders in Canada and the United States. Agencies collect their own flood cost data in ways that are useful to them but may not align with a standardized methodology. Moreover, discrepancies between flood (damage) definitions and flood damage standards

can contribute to wide disparities in disaster information [39]. The damage estimates from different data types make them difficult to compare, as data collections are designed at different scales with different collection standards and methods, indicating the need for better coordination across scales, institutions, and jurisdictions. This consistency across space and time is critical for future analyses of changes in flood damages resulting from global climate change versus local climate variability. Moreover, complicated data sharing codes and data privacy vary from community to community and among multiple provinces or states.

## 5. Conclusions

In this study, we applied the methodology developed by Adeel et al. [1] to major flood events from 2013–2017 and three in-depth case studies across Canada, Mexico, and the United States. In our comprehensive evaluation of data availability for 105 flood damage and loss indicators in the methodology, we found that Mexico had the most complete flood economic impact data, whereas Canada had the least complete data. The structure of Mexico's centralized disaster prevention agency facilitates ease of flood cost data collection and tracking. In contrast, data collection by multiple agencies at multiple scales in the United States and Canada complicates both data coordination and flood damage and loss assessment. In the U.S. case, the lack of a single accounting standard and centralized data collection system between state and federal government sources increases the likelihood of double counting or inaccurate damages and loss estimates. Canada and the United States collect minimal fine-scale data for the majority of flood damage and loss indicators defined in the methodology.

We noted that most economic flood damage assessments focused only on the evaluation of direct damages in the three countries, with a strong emphasis on housing sector direct damages. In contrast, data for indirect effects identified by the methodology were rarely collected in these countries—a key gap in publicly available data collection. Effective and comprehensive economic flood cost assessments must be sustained through mutual collaboration. Our study provides a foundation to bring flood cost data consistency across the three countries, but coordinated efforts (primarily by federal and state/provincial governments and the insurance industry) are needed to build a robust data-management system that enables the countries to plan for, adapt to, and mitigate damages from flooding events, especially in the face of climate change.

**Supplementary Materials:** The following are available online at https://www.mdpi.com/article/10.3390/su142114139/s1, Table S1. Three case studies across Canada, Mexico, and the United States [20,26,40–51]. Table S2. Data processing and analysis across Canada, Mexico, and the United States.

**Author Contributions:** All authors contributed to conceptualization, methodology, and writing—review and editing. Data curation: A.M.A.F. (Mexico flood events and case study), L.M.R. (U.S. flood events and case study), H.S. (Canada flood events and case study), and X.W. (synthesized the three countries' flood events). Formal analysis: A.M.A.F. (Mexico case study), L.M.R. (U.S. case study), H.S. (Canada case study), and X.W. (analysis of three countries' flood events). Writing—original draft preparation: A.M.A.F. (Mexico case study), L.M.R. (U.S. case study), H.S. (Canada case study), and X.W. (rest of original draft preparation and integrated three case studies into the original draft). Visualization: A.M.A.F. (Mexico case study), L.M.R. (U.S. case study), H.S. (Canada case study), and X.W. (rest of visualization/data presentation and adjusted the format and unit of data presentation in three case studies). Supervision: Z.A., L.A.B., K.M.M.E., G.M.G., R.A.M., and E.F.V.; project administration: Z.A.; funding acquisition: Z.A. All authors have read and agreed to the published version of the manuscript.

**Funding:** This study was funded by the Commission for Environmental Cooperation.

**Institutional Review Board Statement:** Not applicable.

**Informed Consent Statement:** Not applicable.

**Data Availability Statement:** Not applicable.

**Acknowledgments:** This work is the product of cooperation of many experts in key institutions in Canada, Mexico, and the United States. We thank the Commission for Environmental Cooperation for its assistance in administration, coordination, and organization of the work. We also thank all workshop attendees, who provided valuable suggestions and comments for the methodology. We would like to thank the reviewers for their valuable comments.

**Conflicts of Interest:** The authors declare no conflict of interest.

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
