# Peer review of "A Comprehensive Methodology for Evaluating the Economic Impacts of Floods: An Application to Canada, Mexico, and the United States"

_sustainability, doi:10.3390/su142114139_

Round 1

Reviewer 1 Report

Dear Authors,

I was invited to review the Manuscript Number sustainability-1931643 -: “A Comprehensive Methodology for Evaluating Economic Impacts of Floods: An Application to Canada, Mexico, and the 3 United States.”.

As a companion paper, this study presents findings from analysis of applying the methodology to investigate economic costs for major flood events between 2013 and 2017 and to assess gaps in the existing data sets across Canada, Mexico, and the United States. I found the general structure of the work well presented and developed. The methodology is well designed and presented. As discussion and conclusion I believe that this manuscript could help to highlight the important to investigate more on collecting information on indirect losses. Few improvements can be added to the introduction and discussion. In particular,

- in paragraph 2.1: I invite to present, further the methodology development, also a short description of the methodology. I understood that is a companion paper of Adeel et al. 2020, but at the same time is also important to help the reader of this paper to have an overview of the methodology without for it to read Adeel or UN-ECLAC. Please also add in this section that the methodology adopted in this manuscript “was largely based on the adapted UN-ECLAC methodology used in Mexico”, as it is written in the discussion session. I invite to clarify better, it is already written between line 111-117, but I believe it will help to understand from the beginning the results regarding the event in Mexico.

- in paragraph 2.2: the selection of the study cases were based also on the criteria of event “well recorded and accessible” (Line 144), this make a strong bias regarding the main aim of this work. This is recognized in the discussion (paragraph 3.4), but I kindly suggest to expand a bit more the discussion on this aspect. Here few questions to suggest the discussion: how will change the results if the selection of the events will be based on different criteria? what are the consequences of selection the sample period (2013-2017)?

Finally, below I suggest couple of pertinent references on the role of indirect losses, they can help to highlight the relevance of them and propose methodology and indicators for indirect loss (e.g., social and economic impacts):

·      “Many studies have highlighted that underestimating indirect losses and additional effects misguides flood risk mitigation and adaptation, as flood indirect impacts may cause significant economic damages [45- 46 – ADD REF 1].”

·      “In addition, methodology data could be coupled with additional geospatial community resilience, flood hazard, and response data to better inform emergency management, mitigation activities, and flood planning that are critical to combat 359 these disastrous events now and into the future [ADD REF 2]”.

To conclude, even if these suggestions are little compare the many aspects present in this work, I believe that these require a major revision of the paper, and if you will consider these revisions, I will be very happy to revise it. 

Suggested references:

1) Arosio M., Arrighi C., Cesarini L., Martina M.L.V., Service Accessibility Risk (SAR) Assessment for Pluvial and Fluvial Floods in an Urban Context, Hydrology, 2021, https://doi.org/10.3390/hydrology8030142

2) Arosio M., Cesarini L., Martina M.L.V., Assessment of the Disaster Resilience of Complex Systems: The Case of the Flood Resilience of a Densely Populated City, Water, 2021, https://doi.org/10.3390/w13202830

Author Response

Dear Reviewer,

We appreciate the time and effort that you dedicated to provide feedback on our manuscript. We are grateful for the insightful comments on and valuable improvements to our manuscript.

We have included a point-by-point response to the reviewer, and please see the attached PDF for details. 

Your sincerely

Reviewer 2 Report

The paper about  A Comprehensive Methodology for Evaluating Economic Impacts of Floods: An Application to Canada, Mexico, and the United States presents findings from analysis of applying the methodology to investigate economic costs for major flood events between 2013 and 2017 and to assess gaps in the existing data sets across.

For publication the authors must made some improvements:

- Please clarify in the introduction part the novelty of the paper in relation to the study conducted by Adeel et al., what is approached differently, what are the new contributions the paper has?  What are the limitations that this paper has solved compared to the methodology proposed by Adeel et al?

- It is not clear what is the gap in the knowledge that is to be addressed by this paper.

- Why was the period 2013-2017 taken into consideration? And why were the same case studies selected as in the previously published article?

- why it is included in the Indigenous perspective analysis is not clearly explained 

- I recommend reorganizing the information as it is hard to get through.

- In figure 2 can be included 3 captures with the area of interest .

- I recommend a correlation between figure 3 and table 3. Possibly a numbering or coding of the events in the realization of figure 3. 

-For better representation I recommend that tables 7-10 be in graphical form, if possible.

-How does the study solve the problem of data collection/availability. Some measures/proposals to improve data collection would be indicated.

Author Response

Dear Reviewer,

We appreciate the time and effort that you dedicated to provide feedback on our manuscript. We are grateful for the insightful comments on and valuable improvements to our manuscript.

We have included a point-by-point response to the reviewer, and please see the attached PDF for details. 

Yours sincerely

Reviewer 3 Report

The authors investigate economic costs for major flood events between 2013 and 2017 in this paper. As we can see in Table 3, the authors summarize major flood events for three countries during 2013 to 2017.  Is there any logistic to select these major flood events? The reason I'm asking is because 2017 Hurricane Harvey caused severe flooding and tremendous economic losses, but I do not see this event included in Table 3 and Section 3.1.

Author Response

(The authors gave the same response as above.)

Round 2

Reviewer 1 Report

Dear Authors,

thank you very much to seriously considered my previous comments and accordingly modified the manuscript.

I have no other comments and I accept it in present form. thank you!

Reviewer 2 Report

Dear author,

Thank you for the punctual response.